# Batched Low-Rank Adaptation of Foundation Models

**Yeming Wen & Swarat Chaudhuri**[*]
Department of Computer Science
The University of Texas at Austin

## Abstract

Low-Rank Adaptation (LoRA) has recently gained attention for fine-tuning foundation models by incorporating trainable low-rank matrices, thereby reducing the number of trainable parameters. While LoRA offers numerous advantages, its applicability for real-time serving to a diverse and global user base is constrained by its incapability to handle multiple task-specific adapters efficiently. This imposes a performance bottleneck in scenarios requiring personalized, task-specific adaptations for each incoming request.

To mitigate this constraint, we introduce Fast LoRA (FLoRA), a framework in which each input example in a minibatch can be associated with its unique low-rank adaptation weights, allowing for efficient batching of heterogeneous requests. We empirically demonstrate that FLoRA retains the performance merits of LoRA, showcasing competitive results on the MultiPL-E code generation benchmark spanning over 8 languages and a multilingual speech recognition task across 6 languages.

## 1 Introduction

Transformer-based foundation models have showcased remarkable performance across various natural language processing tasks, as evidenced by the successes of ChatGPT (OpenAI, 2023), GitHub Copilot (Chen et al., 2021) and Speech Recognition (Radford et al., 2022) among others. The practice of fine-tuning these models for specific domains or specialized needs, such as instruction-tuning, has become increasingly prevalent (Wang et al., 2022c; Honovich et al., 2022; Taori et al., 2023; Chiang et al., 2023). This is driven by the requirements of real-world applications, which often demand models tailored to specific domains, tasks, or even individual user preferences (Ouyang et al., 2022). However, the extensive number of parameters in foundation models poses computational and memory challenges for task-specific fine-tuning.

Low-Rank Adaptation (LoRA) emerged as a solution to this challenge by incorporating trainable low-rank matrices (Hu et al., 2021) which significantly reduces the number of trainable parameters during fine-tuning. LoRA's success stems from its ability to achieve domain adaptation without retraining the entire model (Taori et al., 2023; Dettmers et al., 2023; Lee et al., 2023). However, a practical challenge arises in real-time serving scenarios. Batching is the practice of aggregating multiple data points into a single computation. It is a common technique to leverage parallel processing capabilities in GPUs, ensuring higher throughput and lower serving cost. It becomes especially crucial when serving world-wide users where many requests could flood in every second. The intrinsic design of LoRA dictates that every example within a batch shares the same adapter, which is suboptimal for real-world serving scenarios where each request may require a unique adapter.

Consider a scenario where users from various locations and professions demand different language and occupation adapters as illustrated in Fig. 1. With LoRA, the batch processing would either force all these diverse requests to share the same adapter or process them sequentially, both of which are impractical. These limitations emphasize the need for a solution that can not only utilize the advantages of LoRA but also serve multiple adapters in parallel, catering to the diverse and simultaneous requests encountered in reality.

---

[*]ywen@utexas.edu, swarat@cs.utexas.edu

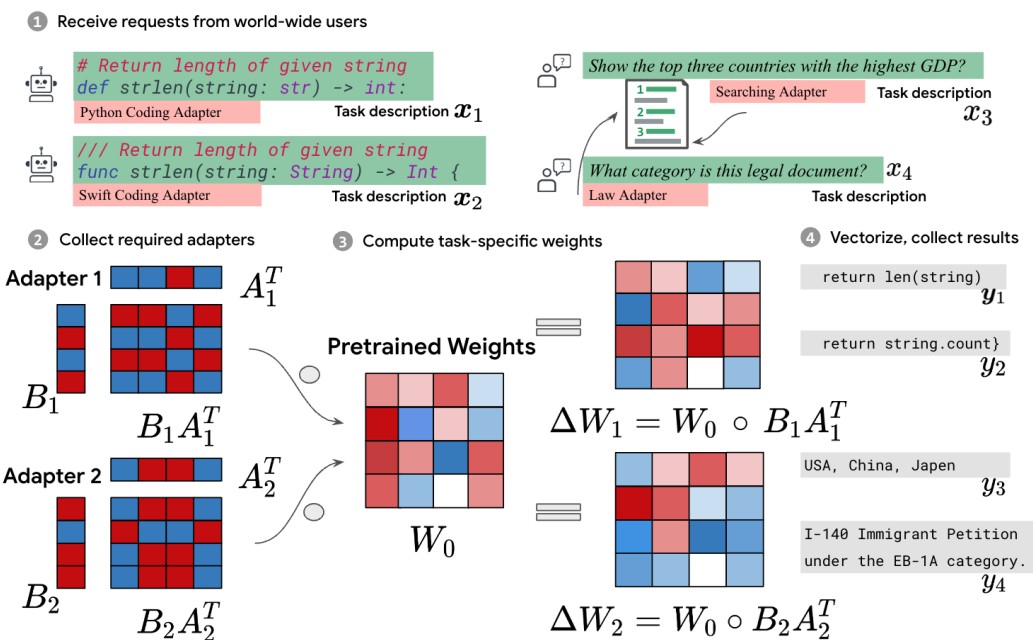

Figure 1: This shows a pragmatic scenario where a foundation model in production receives four incoming requests, each requiring distinct adapters. Omitting two adapters in step 2 & 3 for presentation simplicity. FLORA facilitates batching in such serving circumstances, provided the adapters are of low rank, thereby sustaining high throughput and low latency. Detailed discussion on vectorization is provided in §3.2.

We posit that it is critical to develop a more flexible adaptation mechanism that is compatible with diverse real-world user queries. We introduce fast LoRA (FLORA), a modification of LoRA, enabling individual examples in a minibatch to be associated with *distinct* low-rank weights without compromising the *expressive power*. This modification promises the benefits of domain adaptation, as heralded by LoRA, but without the batching limitation.

Our contributions can be summarized as follows:

1. We propose FLORA, a framework that augments LoRA by allowing each example in a minibatch to have its unique low-rank adapters, facilitating efficient batching.

2. We provided an analytical analysis describing the scenarios where FLORA would be preferred over LoRA in practical applications. This analysis is further substantiated by the empirical evidence where FLORA achieves a 2X throughput improvement on the state-of-the-art code LLM StarCoder 15B in the low-rank setting when diverse adapters are required for incoming examples. Additionally, FLORA reduces the latency by half under the same low-rank setting.

3. We demonstrate that FLORA does not sacrifice accuracy compared to LoRA on a multilingual code generation task across 8 programming languages, and maintains accuracy in speech recognition tasks over five languages.

## 2 PROBLEM FORMULATION

In this section, we outline the problem tackled in this work, illustrating the constraints and objectives that drive the development of the proposed FLORA methodology. Let $\mathcal{M}$ denote a foundation model parameterized by $\theta$, with a total number of parameters $N$. The common practice is to fine-tune this foundational model for various specific tasks, ranging from multilingual code generation to speech recognition as demonstrated in §4.2 and §4.3.

## 2.1 LoRA Adapters

Fine-tuning the entire model $\mathcal{M}$ for a specific task is usually computationally expensive due to the massive parameter count. LoRA (Low-rank Adaptation, (Hu et al., 2021)) was introduced to facilitate domain-specific adaptations with a significantly reduced parameter footprint, with the hypothesis that low-rank adaptation is sufficient for fine-tuning domain specific foundation models.

Given the pre-trained weight matrix $W_0 \in \mathbb{R}^{d \times k}$, LoRA posits that the weight matrix of the adapted foundation model can be expressed as $W_0 + \Delta W = W_0 + BA$, where $\Delta W$ has a low-rank structure. This matrix $\Delta W$ is factorized into two smaller, trainable matrices: $B \in \mathbb{R}^{d \times r}$ and $A \in \mathbb{R}^{r \times k}$, such that $\Delta W = BA$ where $r$ stands for the rank. For a given input $\mathbf{x}_i$, the output $\mathbf{y}_i$ is given by:

$$\mathbf{y}_i = \mathcal{M}(\mathbf{x}_i | \Delta W, W_0, \theta) \tag{1}$$

## 2.2 Batching & throughput

Batching is a common practice where multiple data points $(\mathbf{x}_1, \mathbf{x}_2, \ldots, \mathbf{x}_m)$ are aggregated into a single batch $\mathcal{B}$. Consequently, the forward passes of these data points are processed concurrently rather than individually. This practice leverages the parallel processing capability of a modern GPU, thereby significantly improving the throughput $T$, i.e., the number of data points processed per unit of time. In the context of foundation models, throughput of a batch $\mathcal{B}$ can be defined as $T = \sum_{i=1}^{m} |\mathbf{y}_i| / \Delta t$, where $|\mathbf{y}_i|$ is the number of tokens generated for each example $\mathbf{x}_i$ in the batch, $\Delta t$ is the total time taken to process the batch, and $m$ is the number of examples in the batch. Note that batching incurs minimal latency penalties. However, given its substantial increase in throughput, batching and its variants are widely used in the state-of-the-art foundation models serving framework such as vLLM (Kwon et al., 2023) to achieve the best balance between throughput and latency.

## 2.3 Objective

Batching typically assumes the same model parameters are utilized for every input example within a minibatch. Hence, a straightforward application of batching in LoRA requires that the adapter matrix $\Delta W$ be shared across all inputs in the batch $\mathcal{B}$. The challenge arises when considering a scenario where each input example in the batch might originate from a different task. Sharing the same $\Delta W$ for all $\mathbf{x}_i$ in $\mathcal{B}$ becomes suboptimal where each input potentially demands a unique adapter. The limitation is particularly acute when the model is expected to serve a world-wide user base with diverse incoming requests.

Given the limitations of LoRA in batching, our objective is to maximize the throughput $T$ in global user serving scenarios by maintaining the batching mechanism. Formally, for each $\mathbf{x}_i \in \mathcal{B}$, we aim to compute $\mathbf{y}_i = \mathcal{M}(\mathbf{x}_i | \Delta W_i, W_0, \theta)$, where $\Delta W_i$ is the adapter matrix corresponding to the input example $x_i$. Therefore, $\Delta W_i$ can be unique across $\mathcal{B}$ and specific to a domain or user preference.

# 3 FLORA: FAST LOW RANK ADAPTATION

As shown in §2.3, adapter sharing is often impractical in real-world serving scenarios. The innovation of FLoRA is the introduction of example-specific adapter $\Delta W_i$ for each $\mathbf{x}_i$ in a minibatch. In FLORA, the weight matrix $W_i$ for each example $\mathbf{x}_i$ in the minibatch is calculated as $W_i = \Delta W_i \circ W_0$, where $\circ$ denotes element-wise multiplication, $W_0$ is the pre-trained weight matrix and $\Delta W_i$ is a low-rank adaptation specifically designed for $\mathbf{x}_i$. Similar to Hu et al. (2021), $\Delta W_i$ is decomposed into two trainable matrices: $B_i \in \mathbb{R}^{d \times r}$ and $A_i \in \mathbb{R}^{r \times k}$, such that $\Delta W_i = B_i A_i$, as shown in Fig. 1. Note that FLORA has the same expressive power as LoRA by its construction.

## 3.1 Forward pass

The advantage of FLORA is that computations on a minibatch can be written in terms of matrix multiplications. This enables efficient batched implementations on modern accelerators such as GPUs. Let $x_i$ denote the activations in one layer of a neural net, which is a vertical vector of length $d$. The next layer's activations are given by

$$y_i = \phi(W_i^T x_i) \tag{2}$$

$$= \phi\big((W_0^T \circ \Delta W_i^T)x_i\big) \tag{3}$$

$$= \phi\big((W_0^T \circ (B_i A_i)^T)x_i\big) \tag{4}$$

$$= \phi\Big(A_i \circ \big(W_0^T (B_i \circ x_i)\big)\Big) \tag{5}$$

When the rank is greater than one, we extend the use of the symbol "$\circ$" to denote potential broadcasting. Additionally, a dimension reduction operation such as `torch.mean` is required prior to applying the activation function $\phi$.

The key to FLORA's flexibility lies in the low rank decomposition enables the incorporation of example-specific adapters directly into the forward pass, as demonstrated in the equations above. Crucially, each of these operations—the element-wise multiplication between $A_i$ and $x_i$, and between $B_i$ and $y_i$ — is inherently batch-friendly. Consequently, FLORA allows for simultaneous processing of multiple requests, each requiring its own adapter, within a single minibatch. To vectorize all adapters in the minibatch, we define matrices $\mathbf{A}$ and $\mathbf{B}$ whose rows correspond to the adapters $A_i$ and $B_i$ for all examples in the minibatch. The above equation is vectorized as:

$$\mathbf{Y} = \phi\Big(\mathbf{A} \circ \big((\mathbf{B} \circ \mathbf{X})W_0\big)\Big) \tag{6}$$

## 3.2 COMPUTATIONAL EFFICIENCY

The computational analysis primarily concentrates on the examination of fully connected layers within a transformer architecture, given that LORA is specifically applied to these layers, such as query and key projections. To begin, we analyze a baseline that leverages batch matrix multiplication to facilitate the serving of LORA with multiple adapters. This operation is possible under the assumption that every adapter required by the input examples in the minibatch shares the same shape, specifically, the same rank. The batch matrix multiplication (BMM) can be implemented using the `torch.bmm` operator in deep learning frameworks such as PyTorch (Paszke et al., 2019). Note that the BMM operator is typically unfavorable in practical settings due to the significant overhead it introduces (Abdelfattah et al., 2016). This overhead diminishes the throughput and increases latency, which is detrimental in serving scenarios where response times are crucial for maintaining a good user experience.

Let $b$ and $l$ denote the batch size and the maximum sequence length in the input batch $\mathcal{B}$. Revisiting the notation introduced in §3, where $W_0 \in \mathbb{R}^{d \times k}$, $B_i \in \mathbb{R}^{d \times r}$ and $A_i \in \mathbb{R}^{r \times k}$, the operations required to compute the pre-activation for an input batch $\mathcal{B}$ with dimensions $[b, l, d]$ consist of one matrix multiplication and two BMMs. The matrix multiplication occurs between the input batch $\mathbf{X}$ and the pre-trained weight $W_0$. The two BMM operations are conducted firstly between the input batch $\mathbf{X}$ and $\mathbf{B}$, and secondly between the result of the prior computation and $\mathbf{A}$, where $\mathbf{A}$ and $\mathbf{B}$ are matrices whose rows correspond to the adapters $A_i$ and $B_i$ for all examples in the minibatch respectively. Assuming for simplicity that the layer neither upscales nor downscales the hidden dimension (i.e. $d = k$), the upper bound complexity of this layer is discerned as $2c_1(dblr) + c_2(bld^2)$, with $c_1$ and $c_2$ representing the computational coefficients of BMM and matrix multiplication respectively. Note that $c_1 >> c_2$ because the BMM operator is more expensive than matrix multiplication.

For FLORA, the cost is one matrix multiplication which is $c_2(rbld^2)$ where $r$ denotes the rank of the adapters. We omit the cost of element-wise multiplication in this analysis because it is negligible to the matrix multiplication cost. Comparing the computational cost of FLORA and LORA boils down to the following inequality

$$\frac{2c_1}{dc_2} + \frac{1}{r} \geq 1 \tag{7}$$

FLORA exhibits a lower computational cost than `bmm` LORA whenever the above inequality holds true. The benefit of FLORA over LORA is notably pronounced when $r = 1$. As the rank increases,

LORA gradually becomes less costly. From the established inequality, a variety of scenarios can be inferred where FLORA has an advantage over LORA. Firstly, the advantage of FLORA is significantly apparent when the rank of adapters is small. Secondly, in configurations where the model has fewer hidden units but an increased number of layers, FLORA tends to outperform LORA due to the smaller value of $d$ in the denominator of Eq. (7).

Another advantage of FLORA is the cost remains invariant to the number of adapters required by the input batch. While the preceding analysis assumes that every token in an example $x_i$ shares the same adapter, it is possible to apply multiple adapters to a single example by dividing the example into chunks, and then applying different adapters to each chunk. This approach is commonly observed in the Mixture of Experts framework (Fedus et al., 2021; Lepikhin et al., 2020; Puigcerver et al., 2023). Incorporating several adapters in an input example notably amplifies the ratio $c_1/c_2$ in Eq. (7)[1], thereby significantly increasing LORA's cost.

The ratio $c_1/c_2$ might not be the same across different transformer architectures. §4.1 is designed to provide a deeper insight into how comparative serving efficiency of FLORA and LORA changes under various architectures. Additionally, it's worth noting that LORA does not apply to the self-attention layers, which constitute a non-trivial portion of the computational cost, thereby overshadowing the advantage of FLORA. However, as efficient self-attention mechanisms such as flash attention (Dao et al., 2022) get adopted, the advantage of FLORA over LORA is likely to get larger.

**Connection to IA3.** IA3 was proposed in Liu et al. (2022a) featuring fast adaptation of LLM. It introduces a learned vector $l$ which re-scales the activation by $y_i = l \circ \phi(W_0^T x_i) = \phi(l \circ (W_0^T x_i))$. This can be viewed as a special case of FLORA – a rank 0.5 variant — which only re-scales the columns instead of the entire pre-trained weights. It has a limited expressive power compared to FLORA and LORA.

## 4 EXPERIMENTS

In this section, we compare FLORA to LORA and other notable baselines across various metrics and tasks. To begin with, we delve into a computational analysis to substantiate the enhanced throughput and the reduced latency achieved by FLORA in the case of low rank Subsequently, we pivot towards analyzing the accuracy of FLORA in multilingual code generation tasks spanning across different languages. The goal is to discern the proficiency of FLORA in maintaining, if not enhancing, the model's accuracy as compared to LORA. Progressing further, we replicate a similar analysis but in the domain of multilingual speech recognition.

### 4.1 SERVING ANALYSIS

The primary objective of this serving analysis is to measure the maximum throughput both FLORA and LORA can attain under varied rank configurations. We carried out this exploration on the state-of-the-art code Large Language Model (LLM) StarCoder (Li et al., 2023), evaluating models of different number of parameters namely 1B, 3B, and 15B. The dataset facilitating this analysis has been sourced from the vLLM throughput benchmark. Noteworthily, this dataset was previously used to fine-tune the English Vicuna model, a state-of-the-art chat LLMs (Chiang et al., 2023). To expedite the benchmarking process, we extracted a subset of 1,000 samples from the original dataset, ensuring a diverse range of sample lengths varying from 50 to 2,000 tokens.

In setting up the computational analysis, our primary intent is to compare FLORA and LORA in the real-world serving scenario. See Appendix B.1 on how bmm LORA is implemented. The vLLM framework (Kwon et al., 2023)[2], with its implementation of continuous batching, presents an ideal setup for this analysis. The continuous batching mechanism in vLLM, as inspired by the principles delineated in Orca (Yu & Jeong, 2022), facilitates a more efficient utilization of GPU resource by allowing new sequence to be inserted immediately once any sequence in the current batch is completed. This continuous flow significantly enhances GPU utilization as compared to static batching, where the GPU awaits the completion of all sequences in a batch before initiating a new batch processing. The comparison of FLORA and LORA within this setup offers a compelling evidence of

---

[1]The batch size in BMM operator increases from $b$ to $b \times m$ where $m$ is the number of adapters per example.
[2]The version is 0.1.3.

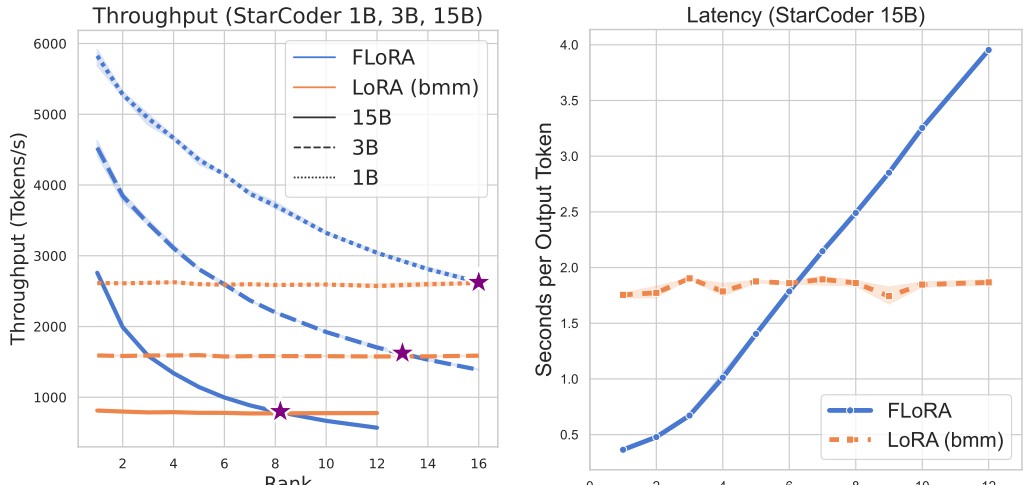

Figure 2: **Left**: Generation throughput vs. rank for FLoRA and `torch.bmm` implementation of LoRA, measured in tokens per second (token/s). The experiments were conducted on three star-coder models: StarCoder 15B, StarCoderbase 3B and StarCoderbase 1B. FLoRA has great through-put advantage over LoRA when the rank is small. As the rank increases, the `torch.bmm` imple-mentation of LoRA finally has a better throughput. **Right**: Latency vs. rank on StarCoder-15B. Requests are coming at the speed of 8 requests per second.

their respective throughput and latency in the real-world serving scenario. It is worth noting that the experiments were conducted without Flash-attention (Dao et al., 2022).

**Throughput experiment.** In Fig. 2, the throughput results for both FLoRA and bmm LoRA are illustrated across different rank configurations on three StarCoder models with different number of parameters, namely 1B, 3B and 15B. All experiments were conducted on an NVIDIA H100 GPU with a `float16` precision[3]. The maximum number of batched tokens is 8,192, which is the same as the model context length. Evidently, FLoRA shows a superior throughput over LoRA in the lower rank configurations. At rank 1, the throughput of FLoRA is more than threefold higher than that of LoRA, thereby highlighting the considerable serving performance boost FLoRA provides. The advantage of FLoRA continues as the rank increases, albeit with a diminishing rate. For instance, at rank 2, FLoRA's throughput is around 2.5 times higher, and this multiplicative advantage decreases as the rank increases further. This performance advantage continues up until the rank of 8 in the StarCoder 15B model, where LoRA starts to outperform FLoRA. This inflection point suggests that the advantages of FLoRA in terms of throughput are more pronounced in lower rank.

Notice that the inflection points occurred at a higher rank when serving a smaller LLM as illustrated in (Fig. 2, **left**). This demonstrates a significant potential of FLoRA, especially when considering future applications of quantization techniques to serving LLMs. By applying quantization, such as 8-bit or 4-bit inference, the effective size of the model is reduced, akin to serving a smaller LLM thus potentially extending the rank at which FLoRA maintains a throughput advantage over LoRA.

**Latency experiment.** We assessed the latency-versus-rank performance on the Starcoder 15B model, using the same dataset as the throughput experiment. This evaluation was conducted un-der the conditions where requests arrive at a rate of 8 per second. The default maximum number of batched tokens in vLLM serving launcher is 2,560. The results, as shown in (Fig. 2, **right**), measure latency in terms of seconds per output token. Remarkably, in the lower rank regime (ranging from rank 1 to rank 4), FLoRA exhibited a 2-5X reduction in latency compared to LoRA. Notably, the latency for LoRA stood at approximately 2.3 seconds per output token, is impractical for serving due to the poor user experience it would cause. This experiment further highlights the superior capabilities of FLoRA in efficiently catering to diverse incoming user requests.

These findings validate the theoretical analysis in §4.1, confirming that FLoRA provides significant throughput advantages, particularly in settings with lower to moderate ranks. This positions FLoRA

---

[3]The quantization in vLLM is still under development.

| Language | pass@1 (Relative Improvement) | | | |
|---|---|---|---|---|
| | Base Model | FLoRA | IA3 | LoRA |
| **StarCoder 15B** | | | | |
| Dlang | 14.13 | 17.26 (22.14%) | 15.26 (7.99%) | 17.15 (21.37%) |
| Perl | 17.05 | 21.44 (25.76%) | 17.71 (3.90%) | 21.46 (25.76%) |
| Ruby | 1.39 | 24.94 (1692.86%) | 20.80 (1394.64%) | 23.76 (1608.04%) |
| Rust | 22.40 | 26.24 (17.14%) | 23.53 (5.04%) | 26.87 (19.95%) |
| Racket | 10.20 | 12.41 (21.61%) | 11.53 (12.96%) | 12.51 (22.58%) |
| Swift | 16.91 | 20.38 (20.51%) | 18.13 (7.19%) | 20.35 (20.36%) |
| **StarCoderBase 3B** | | | | |
| Dlang | 5.65 | 5.72 (1.20%) | 5.72 (1.20%) | 6.97 (23.34%) |
| Perl | 10.73 | 13.01 (21.25%) | 11.46 (6.83%) | 13.31 (27.51%) |
| Ruby | 5.33 | 14.48 (171.68%) | 7.88 (47.90%) | 13.89 (160.68%) |
| Rust | 17.18 | 21.00 (22.24%) | 17.28 (0.60%) | 20.67 (20.31%) |
| Racket | 8.04 | 9.16 (13.99%) | 8.40 (4.48%) | 8.80 (9.48%) |
| Swift | 10.04 | 15.69 (56.21%) | 12.54 (24.83%) | 15.04 (49.76%) |

Table 1: Comparison of three fine-tuning methods FLoRA, IA3, and LoRA on StarCoder 15B and StarCoderBase 3B across various low-resource programming languages in the MultiPL-E benchmark. The table presents Pass@1 accuracy of each method alongside the relative improvement over the baseline. The standard errors are less than 0.3% in all cells in the table, therefore we exclude that for clear presentation.

as a compelling alternative for efficiently serving adapted foundation models, especially in scenarios where lower ranks suffice the desired model accuracy, as further demonstrated in the subsequent accuracy analysis sections. Moreoever, if an enterprise chooses to serve foundation models with a substantial number of diverse adapters, for instance, a personalized LLM, then a low rank or even a rank one is imperative to avoid excessive storage costs.

## 4.2 MULTILINGUAL CODE GENERATION

A key aspect to examine before applying FLoRA in real-world LLM serving is to scrutinize any potential degradation in performance. In this section, we consider multilingual code generation as the testbed for comparing FLoRA and LoRA due to its alignment with real-world applications, where the necessity to cater to diverse programming languages is paramount. Low-resource languages, as referred to in this context, are languages that appear much less frequently than other languages in the pre-training data. Orlanski et al. (2023) showed that the performance of code LLMs can be notably enhanced on low-resource programming languages such as PERL by recalibrating the language distribution in the pre-training data. This suggests that fine-tuning a trained LLM on such low-resource languages could potentially boost its performance on the same language. Hence, by employing multilingual code generation as a benchmark, we can make an informed evaluation of adaptability and performance enhancements that FLoRA and LoRA can achieve.

Additionally, a comparison is made against a third baseline, IA3, which can be considered as a special case of FLoRA. Essentially, IA3 can be seen as a rank 0.5 variant of FLoRA, thereby facing a constrained expressive power in comparison to both FLoRA and LoRA. For FLoRA and LoRA, we conducted fine-tuning across a range of rank choices, spanning from 1 to 8. It emerges that within the scope of this multilingual code generation task, a rank of one typically suffices to achieve optimal results, with the exception of Racket and Lua. Consequently, the results shown in §4.2 are predominantly at rank 1, barring Racket and Lua, which are presented at rank 4.

**Fine-tuning.** In our effort to evaluate the performance of FLoRA, LoRA, and IA3 on the multilingual code generation task, we fine-tuned these models on state-of-the-art multilingual code LLMs, StarCoder 15B and StarCoderBase 3B as introduced in (Li et al., 2023). A pivotal aspect of our fine-tuning process was the utilization of existing data, negating the need for creating new data for

| Model | Arabic | Czech | Lithuanian | Marathi | Mongolian | Hindi |
|---|---|---|---|---|---|---|
| Whisper | 46.03 | 23.19 | 46.09 | 84.84 | 115.20 | 43.41 |
| FLoRA | 30.21 ± 0.25 | 10.76 ± 0.22 | 20.39 ± 0.32 | 33.17 ± 0.17 | 42.12 ± 0.63 | 23.58 ± 0.65 |
| IA3 | 30.48 ± 0.22 | 11.61 ± 0.33 | 21.41 ± 0.20 | 35.03 ± 0.61 | 46.47 ± 0.53 | 25.64 ± 0.78 |
| LoRA | 30.18 ± 0.21 | 10.77 ± 0.38 | 20.50 ± 0.21 | 31.94 ± 0.33 | 41.57 ± 0.69 | 23.82 ± 0.74 |

Table 2: Mean WER with standard deviation for different models and languages. The WER is calculated with respect to the unnormalized tokenizer. The base model here is Whisper-1.5B.

low-resource programming languages. We leveraged the same pre-training data that was used for pre-training StarCoder, specifically, the Stack dataset, which contains over 6TB of permissively-licensed source code files covering 358 programming languages. For each low-resource language in our experiment, we fine-tuned on its corresponding split from the Stack dataset for a total of 1,500 steps, along with batch size 8. More fine-tuning details are given in Appendix A.

**Evaluation.** The evaluation of FLoRA, LoRA, and IA3 was conducted on the MultiPL-E benchmark (Cassano et al., 2023), which contains the translation of two unit-test driven Python benchmarks (HumanEval and MBPP) to 18 other programming languages. We used the HumanEval split of the benchmark to evaluate the fine-tuned models. As for the metrics, We adopted the $pass@k$ metrics from Chen et al. (2021); Austin et al. (2021), which is calculated as the fraction of problems with at least one correct sample given $k$ samples. Similar to Chen et al. (2021), we drew 100 samples for computing $pass@1$, with a sampling temperature set at 0.1.

**Main results.** The result in §4.2 exhibits the performance of three methods across various programming languages on both StarCoder 15B and StarCoderBase 3B models. The average relative improvement achieved by FLoRA and LoRA is roughly 20% in the selected low-resource programming languages. FLoRA consistently outperforms IA3 on all languages, especially on StarCoder 15B, denoting its efficiency in leveraging the model expressive power to improve multilingual code generation. It is notable that StarCoder 15B has an unforeseen issue regarding Ruby generation, where it yields lower accuracy compared to the 3B model. However, all methods are able to fix its abnormal performance.

On StarCoderBase 3B, a smaller model, it is evident that the baseline performance drops, yet FLoRA and LoRA still manage to exhibit substantial relative improvement over the baseline, especially in languages like Swift and Ruby. This suggests that both methods benefit from continuous training on the low-resource language split of the pre-training data, although the advantages may diminish with a reduction in model size. While the absolute performance ($pass@1$ accuracy) varies among languages, the relative improvements highlight the effectiveness of the tested methods in enhancing multilingual code generation.

### 4.3 MULTILINGUAL SPEECH RECOGNITION

Following our analysis in multilingual code generation, we shift our focus to another substantial application of large language models — multilingual speech recognition (MSR). This domain has seen increasing demand due to the necessity of serving various linguistic interfaces. The capabilities of FLoRA and LoRA to adapt to multiple languages in code generation presents a tempting premise to investigate their effectiveness in the MSR domain.

**Fine-tuning.** A crucial aspect of our analysis involves fine-tuning the foundation model on a multilingual speech recognition dataset. We selected the Whisper large 1.5B (Radford et al., 2022) as our base model to conduct the subsequent analysis. Whisper is an encoder-decoder transformer model trained on 680k hours of labeled speech data through large-scale weak supervision. Despite being pre-trained on a multilingual speech corpus, the model's predictive capabilities can be improved further for certain languages and tasks through fine-tuning. We use the Common Voice benchmark (Ardila et al., 2020) — containing a total of 38 languages and 2,500 hours of collected audio — for the fine-tuning process, with a particular focus on low-resource languages.[4] For each low-resource language enumerated in §4.2, we fine-tuned on its training split within the Common

---

[4]Dataset statistics can be found here https://commonvoice.mozilla.org/en/datasets.

Voice dataset for approximately 5,000 steps, with a batch size of 6. More fine-tuning details are given in Appendix A. Similar to §4.2, we found that rank one is sufficient to achieve the best result in most cases, with the exception of Marathi, which requires a rank of 4.

**Main results.** We present the main results on the MSR task in §4.2. Notice that the evaluation is conducted with the unnormalized tokenizer.[5] The Whisper model exhibited a relatively higher Word Error Rate (WER) across all low-resource languages when compared to FLORA, LORA and IA3 models, reflecting a significant room for enhancement in its multilingual speech recognition capabilities. Particularly, its performance was found to be subpar for the Marathi and Mongolian languages, with WERs of 84.84 and 115.20 respectively. All examined models significantly outperform the base Whisper model by a wide margin across all languages, highlighting the effective fine-tuning of FLORA, LORA and IA3 models. For instance, in Arabic, FLORA reduces the WER to 30.21 from Whisper's 46.03, showcasing a significant enhancement in speech recognition accuracy. Particularly, FLORA and LORA consistently outperform IA3, indicating a better expressive power for multilingual speech recognition tasks. For example, in Czech, FLORA has a mean WER of 10.76, which is slightly better compared to IA3's 11.61 and LORA's 11.47. A similar trend is observed in Lithuanian and Hindi. In summary, this table shows the superior performance of FLORA and LORA over IA3 in the speech recognition task across diverse languages. The consistent low WERs attained by FLORA demonstrates its potential as a viable model for MSR tasks.

## 5 RELATED WORK

Parameter-Efficient Fine-Tuning (PEFT) methods are broadly partitioned into two categories: weight-based and non-weight-based approaches. MC-dropout (Lakshminarayanan et al., 2016) stands as an early example for the non-weight-based approach, where distinct dropout masks are allocated to various tasks. Recently, prompt tuning techniques have emerged as a prevalent stream within this category (Li & Liang, 2021; Lester et al., 2021), facilitating efficient adaptation with minimal modifications to models. Successive endeavors aimed to enhance this class of methods, delving into aspects such as optimization (Mao et al., 2021; Diao et al., 2022), transferability (Wang et al., 2021; Vu et al., 2021; He et al., 2022b), and the usage of discrete prompts (Schick & Schütze, 2020a;b; Gao et al., 2021; Malkin et al., 2021), among others (Liu et al., 2022b; 2021).

We focus on weight-based approaches in this work, which has a weight interpretation as exemplified by LORA (Hu et al., 2021). This line of research can be traced back to Progressive Network (Rusu et al., 2016), which inserts a sub-network when a new task arrives. This principle was later adapted widely in foundation models as represented by adapter based methods (Houlsby et al., 2019; Mahabadi et al., 2021; Davison, 2021; Ding et al., 2022; Wang et al., 2022b). In particular, BitFit (Ben-Zaken et al., 2021) was introduced to solely update the bias parameters, while IA3 (Liu et al., 2022a) was proposed to rescale the activations. Additionally, approaches such Fish (Sung et al., 2021) and Diff-pruning (Guo et al., 2020) leverage sparsity to facilitate efficient adaptation of foundation models. A separate vein of research aims to improve LORA by reducing its computational and memory costs (Zhang et al., 2023b;a; Malladi et al., 2023). He et al. (2022a) explored how to unify different PEFT methods. Dettmers et al. (2023) quantized LORA to reduce memory footprint. Chavan et al. (2023) generalized LORA by learning individual adapters in each layer. Several other works focus on building mixture of adapters (Wang et al., 2022a; Diao et al., 2023).

## 6 CONCLUSION

We introduced FLORA, an extension of LORA, facilitating efficient batching. Empirical evaluations demonstrated that FLORA enhances throughput and latency in practical serving scenarios, all while preserving the accuracy of LORA. Through FLORA, we aim to facilitate a more efficient adaptation of large language models to diverse and real-world user requests.

**Limitations.** Despite its parameter efficiency, FLORA still requires fine-tuning. A promising future work could be to derive FLORA weights from a trained LORA model, given that LORA remains

---

[5]This is because the normalized tokenizer has a bug in the official leaderboard implementation at the time of submission.

the most prevalent type of adapter as per (Huang et al., 2023). This adaptation could potentially obviate the requirement for fine-tuning, thereby accelerating the process of model adaptation.

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

## A  ADDITIONAL TRAINING DETAILS

We delve into additional details regarding the models used in §4.2. We conducted fine-tuning on two StarCoder models (Li et al., 2023), specifically the 15B and 3B models. For both models, we employed LORA (Hu et al., 2021) on the default layers within the Parameter Efficient Fine-Tuning (PEFT) library[6], targeting only the key and query projections in the self-attention layers. This incurs roughly $0.03\%$ number of training parameters in the rank 1 setting. For a fair comparison, we applied FLORA to the same layers where LORA was applied. Regarding the other baseline IA3 (Liu et al., 2022a), we executed the fine-tuning in two distinct settings: one with the default configuration in PEFT, applying IA3 to the key projection layers and the first fully-connected layer in the feed-forward layers; and another aligning with the LORA setting, restricting application to the key and query projection layers. The latter setup proved to be less effective than the default setting, thus the results presented in §4.2 adhere to the default configuration.

Despite StarCoder's pre-training on the Stack dataset, which contains over 6TB of source code files covering 358 programming languages, we still fine-tuned all methods on this dataset for the low-resource programming languages. This eliminates the time-consuming process to create new data. Our observations revealed that both IA3 and FLORA require a higher learning rate compared to

---

[6]https://github.com/huggingface/peft/tree/main

| Language | pass@1 (Relative Improvement) | | | |
|---|---|---|---|---|
| | Base Model | FLoRA | IA3 | LoRA |
| **StarCoder 15B** | | | | |
| Julia | 23.31 | 26.56 (13.92%) | 23.90 (2.51%) | 25.62 (10.45%) |
| Lua | 26.60 | 30.20 (13.57%) | 28.40 (6.77%) | 29.04 (9.18%) |
| R | 16.07 | 17.55 (9.24%) | 16.65 (3.59%) | 18.60 (15.8%) |
| **StarCoderBase 3B** | | | | |
| Julia | 15.62 | 19.11 (22.39%) | 16.72 (7.05%) | 19.77 (26.59%) |
| Lua | 17.63 | 19.46 (10.39%) | 18.04 (2.33%) | 20.23 (14.75%) |
| R | 10.00 | 10.70 (7.02%) | 9.98 (-0.25%) | 10.06 (0.62%) |

Table 3: Comparison of three fine-tuning methods FLoRA, IA3, and LoRA on StarCoder 15B and StarCoderBase 3B across additional low-resource programming languages in the MultiPL-E benchmark.

LoRA. For LoRA, a learning rate of $1e{-}4$ sufficed, whereas IA3 required $5e{-}3$, and FLoRA demanded an even higher rate of $8e{-}3$. We conjecture that this stems from FLoRA and IA3 incorporating multiplicative weight perturbation, which typically requires a larger learning rate. All models were fine-tuned using 8 bit quantization featuring lower training memory cost.

We present results for additional low-resource languages in Appendix A. The improvements provided by PEFT methods are less pronounced in these languages compared to those documented in §4.2. Nevertheless, it can be observed that LoRA and FLoRA still outperform IA3, despite the limited room for enhancement in these languages.

Next, we discuss further details concerning the models used in §4.3. We performed fine-tuning on the Whisper large model (Radford et al., 2022), which consists of 1.5 billion parameters. Unlike the code generation task, we discovered that using the default modules in the PEFT library does not yield the best performance. Instead, we applied FLoRA, LoRA, and IA3 to every fully-connected layer in the Whisper transformer. We fine-tuned all methods on the Common Voice benchmark (Ardila et al., 2020) for each low-resource language showcased in §4.2. We continued to employ larger learning rates for FLoRA and IA3. Specifically, we maintained a learning rate of $2e{-}4$ for LoRA, while using $4e{-}3$ and $6e{-}3$ for IA3 and FLoRA respectively. Additionally, we utilized 8-bit quantization during training to minimize the GPU memory requirement.

## B ADDITIONAL SERVING RESULTS

### B.1 BMM IMPLEMENTATION

We provide further details concerning the experiments in §4.1. To begin, we illustrate the implementation of "`torch.bmm`" as follows. Upon computing the `adaptersout`, it can be added to the output of the standard LLM layer. This method facilitates the computation of diverse adapters' outputs within a batch.

```
inputs = torch.randn(batch, seq_length, d_model)
adapter_b # shape of (batch, d_model, rank)
adapter_a # shape of (batch, rank, d_model)
hidden = torch.bmm(inputs, adapter_b)
adaptersout = torch.bmm(hidden, adapter_a)
```

### B.2 ADDITIONAL RESULTS

We further conducted a throughput experiment, employing a setup similar to the one described in §4.1, on the newly proposed Llama-2 foundation models (Touvron et al., 2023). The results for both the 13B and 7B models are illustrated in Fig. 3 (**left**). When compared against the plot in

Fig. 2 (**left**), it is evident that the inflection rank at which LoRA begins to outperform FLoRA is lower for the Llama-2 models than for the StarCoder 15B. This shows that the ratio $c_1/c_2$ from Eq. (7) varies across different architectures. One plausible explanation could be that, being a recent architecture, the Llama-2 model has not yet been fully optimized within the vLLM framework, potentially impacting throughput performance.

The right panel of Fig. 3 presents the serving results concerning StarCoder-3B. Notably, the inflection rank at which LoRA begins to outperform FLoRA is lower here as compared to the one observed in StarCoder 15B, as seen in Fig. 2 (**right**). This is attributed to the relatively low latency on StarCoder-3B, which implies that a request rate of 8 requests per second fails to saturate the LLM server, thereby diminishing the advantage of FLoRA. Upon escalating the request rate to 15 requests per second, the inflection rank correspondingly increases to around 12. This illustrates that the latency improvement exhibited in Fig. 2 (**left**) represents a theoretical enhancement. The actual improvement in serving depends on model architectures, request rates, and other practical considerations.

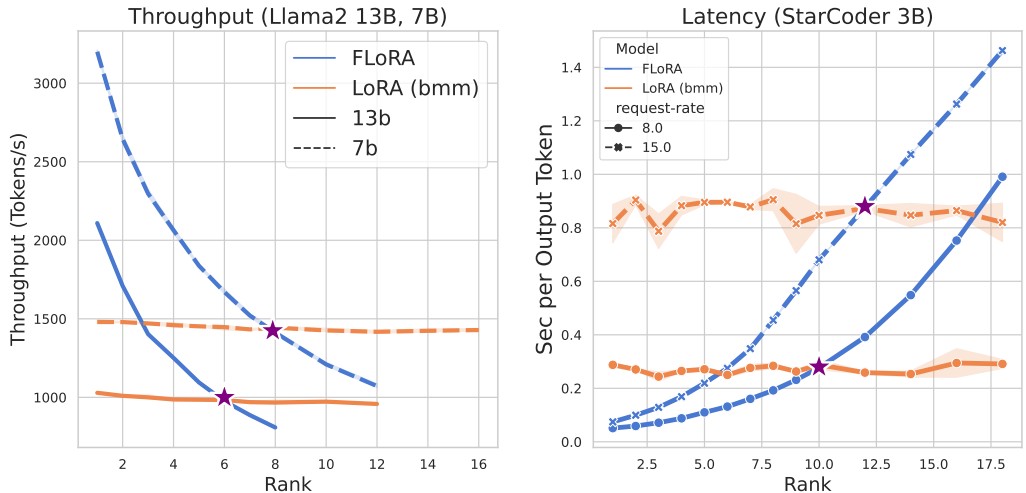

Figure 3: **Left**: Generation throughput vs. rank for FLoRA and `torch.bmm` implementation of LoRA, measured in tokens per second (token/s). The experiments were conducted on two Llama-2 models: 13B and 7B (Touvron et al., 2023). FLoRA has great throughput advantage over LoRA when the rank is small. As the rank increases, the `torch.bmm` implementation of LoRA finally has a better throughput. **Right**: Latency vs. rank on StarCoder-3B. Requests are coming at the speed of 8 requests per second and 15 requests per second.

