# OpenReview forum: "Batched Low-Rank Adaptation of Foundation Models"
_ICLR.cc/2024/Conference — ICLR 2024 oral_

### Official Review · Reviewer_daKT · 2023-10-25

**Soundness:** 4 excellent
**Presentation:** 3 good
**Contribution:** 4 excellent
**Rating:** 8
**Confidence:** 4

**Summary:**

LoRA, a widely used technique for fine-tuning a small number of parameters in foundation models, exhibits a weakness in batched inference settings where each request in the batch requires a unique adapter.
In such a scenario, batched inference using LoRA becomes sequential and inefficient. This paper proposes a variant of LoRA, called fast LoRA (FLoRA), which utilizes a parameterization that enables minibatch computations to be performed using matrix multiplications. This makes it efficient to perform batched inferences with distinct adapters per request.
The paper presents a computational analysis demonstrating that FLoRA can achieve improvements in both throughput and latency compared to LoRA for scenarios involving low-rank and small model dimensions.
The paper presents empirical results demonstrating the advantages of FLoRA over LoRA when using StarCoder (Li et al., 2023) as the foundation model. On multilingual code generation and speech recognition tasks, FLoRA achieves similar performance to LoRA and outperforms IA3.

**Strengths:**

* Proposes an alternative to the LoRA approach that is efficient for batched inference with distinct adapters per request.
* Presents an analysis demonstrating the conditions under which the proposed approach can outperform LoRA.
* Demonstrates using the StarCoder 15B LLM that FLoRA can double the throughput (halve the latency) in a low-rank setting when diverse adapters are required for incoming examples.
* Shows that FLoRA yields similar results as LoRA on multilingual code generation and speech recognition tasks.

**Weaknesses:**

* Some parts of the paper are not clear (see comments below).

**Questions:**

* The transition from Eqn 4 to 5 is not immediately clear. It would be helpful to provide intermediate steps.
* P5: In the sentence, "Secondly, in configurations where the model has fewer hidden units but an increased number of layers, FLORA tends to outperform LORA due to the smaller value of d in the denominator of Eq. (7)." How is the increased number of layers important given that Eq (7) contains only the dimensionality of the hidden units d and the rank r?
* Table 2: Is there any reason why FLoRA underperforms LoRA for Marathi? What is the amount of fine-tuning data for each language?

---

> ### Author Response · Authors · 2023-11-16
>
> Thank you for your insightful review.
>
> > Transition from Eq 4 to Eq 5
>
> We will include more intermediate steps such as the transformation from vector to matrix operators  in the revision.
>
> > How is the increased number of layers important given that Eq (7) contains only the dimensionality of the hidden units d and the rank r?
>
> Thank you for pointing this out. We assume the number of parameters of a LLM is fixed, then a deeper LLM would indicate less number of hidden units. This helps Inequality. 7 hold. We will make this clearer in the revision.
>
> > Is there any reason why FLoRA underperforms LoRA for Marathi? What is the amount of fine-tuning data for each language?
>
>
> The statistics for each language can be found here https://commonvoice.mozilla.org/en/datasets. Each language does have a different number of hours of fine-tuning data. For Marathi, we observed instability in the training loss of fLoRA, which may have contributed to its relative underperformance.

---

> > ### Comment · Reviewer_daKT · 2023-11-21
> > **Thanks for the clarifications**
> >
> > Thanks for adding the clarifications and adding the link to the CommonVoice mozilla corpus.

---

### Official Review · Reviewer_46uC · 2023-10-31

**Soundness:** 3 good
**Presentation:** 3 good
**Contribution:** 3 good
**Rating:** 8
**Confidence:** 3

**Summary:**

The paper proposes FLoRA, which allows each example in a minibatch to own unique low-rank adapters. FLoRA encourages efficient batching of serving various requests, retaining performances of LoRA with throughput improvement and latency reduction in low-rank settings.

**Strengths:**

1. The orientation is clear. It can important to equip language models with various task-specific adapters for diverse requests. The overall idea is well-motivated.
2. The formulation is clear and analysis of computational consumption is in detailed.

**Weaknesses:**

1. If each example in a minibatch has its own adapters, the overall performance is expected to overcome LoRA, however, it's almost the same as LoRA. So the "performance bottleneck in scenarios requiring personalized, task-specific adaptations for each incoming request" isn't largely solved.
2. The whole mechanism and the algorithm isn't mentioned clearly. e.g., how to choose the batch size for real situations, how to make each example corresponding to its appropriate adapters during inference. The paper over-concentrates on Fomulation and Computational Efficiency, while the high-level algorithm--the whole process is not quite clear.

**Questions:**

1. What's the memory comsumption of FLoRA compared with other methods?
2. Can you further explain "FLORA has the same expressive power as LORA by its construction"?
3. The reason for changing "addition" of low-rank adapters in LoRA to "multiplication" in FLoRA is only for computational efficiency or for something else?

---

> ### Author Response · Authors · 2023-11-16
>
> Thank you for your insightful review and the opportunity to address your concerns.
>
> > The "performance bottleneck in scenarios requiring personalized, task-specific adaptations for each incoming request" isn't largely solved.
>
> The primary performance bottleneck we aim to address with fLoRA is computational efficiency, particularly in serving scenarios requiring fast response times. For example, as shown in Fig. 2b, waiting 2.2 sec per output token is impractical in real-world applications. fLoRA provided an alternative solution in such scenarios, improving throughput and reducing latency without compromising accuracy. We will clarify this in our revised manuscript to emphasize that the bottleneck addressed is in computational performance.
>
> > The whole mechanism and the algorithm isn't mentioned clearly. e.g., how to choose the batch size for real situations, how to make each example corresponding to its appropriate adapters during inference.
>
> Thank you for pointing this out. We will make sure to include more algorithmic details in the revision. For the experiment in Sec 4.1, the vLLM framework does not use batch size. Instead, it uses max number batched tokens to control how many tokens can be batched in one forward pass. For the throughput experiment, we set max_num_batched_tokens to be 8192, following the convention of using higher batch size for throughput benchmarking. Assuming the average input length of the testing dataset (the chat dataset to fine-tune the Vicuna model) is 512 then the batch size would be 16. For the latency experiment, we set max_num_batched_tokens to be 2560 which is the default setting of the vLLM framework for online serving.
>
> We assume the simplest case in the serving experiment where all possible adapters have been loaded in the memory. We also assume that each example (each request) in the batch is associated with an adapter id, then the adapter can be chosen by calling torch indexing.
>
> > Memory consumption.
>
> In the serving scenario, we did not observe any memory difference between fLoRA and LoRA. The vLLM framework will fit as many tokens as possible (but less than max_num_batched_tokens), utilizing all GPU memory. In the fine-tuning stage, when the rank is small, the memory consumption of fLoRA is roughly the same as LoRA. However, when the rank gets bigger (say rank > 7), the memory consumption of fLoRA is larger than LoRA. Notice that the self-attention layer accounts for the major memory consumption which fLoRA and LoRA don't touch. We will clarify this in the revision.
>
>
> > What is “same expressive power”
>
> We use "expressive power" to denote the rank of the modulation matrix. Both fLoRA and LoRA have the capability to modulate the weight matrix up to any desired rank. In this perspective, they have the same expressive power. We will make this clearer in the revision.
>
> > Rationale for Multiplication in fLoRA
>
> The transition from addition to multiplication is primarily driven by computational efficiency. The multiplication operator enables the batching mechanism.

---

> > ### Comment · Reviewer_46uC · 2023-11-21
> > **Thanks for the response**
> >
> > I appreciate your response and I'm looking forward to the final version. I will raise my score to 8.

---

### Official Review · Reviewer_Uf9f · 2023-10-31

**Soundness:** 3 good
**Presentation:** 3 good
**Contribution:** 3 good
**Rating:** 8
**Confidence:** 2

**Summary:**

The paper builds up on the Low-Rank Adaptation (LoRA) framework to fine-tune foundation models, by introducing fLoRA, which allows distinct adapters for different task-specific requests within the same batch. The authors empirically demonstrate that their approach preserves the advantages of LoRA in terms of accuracy on multilingual code generation and speech recognition tasks, while facilitating a higher throughput and lower latency.

**Strengths:**

The paper clearly introduces the problem and the contributions compared to the state of the art. The contribution is significant to cope with practical challenges of using foundation models in real-time serving scenarios, especially when considering world-wide incoming requests.
The paper looks theoretically and technically sound and the presentation is clear, well framed in the context, and easy to follow.

**Weaknesses:**

I don’t find major weaknesses. Minor comments are indicated in the following section.

**Questions:**

-	I suggest removing references from the abstract.
-	Could you explicitly clarify the definition of “expressive power” in the paper?
-	About contribution 3 (Introduction): since fLoRA allows task-specific adapters for fine-tuning, wouldn’t you expect a higher, rather than simply equivalent, accuracy compared to fLoRA? In which scenarios do you expect that fLoRA could have sacrificed accuracy compared to LoRA?
-	Fig 1: The figure is useful, but framing the different sections (1,2,3,4), or at least avoid overlapping among them would help clarity. Also, 4 task descriptions are indicated at point 1, and the corresponding 4 results are shown at point 4, while only 2 adapters and weights computations are shown at point 2 and 3. In my view, it would be clearer to keep the number of examples consistent across the sub-figures.
-	In LoRA the weight matrix of the adapted foundation model is expressed by the SUM of W0 and DeltaW, while in fLoRA the weight matrix specific for each example is calculated as the element-wise MULTIPLICATION of W0 and DeltaWi. Is this correct?
-	On paragraph 3.2 you say that “fLoRA exhibits a lower computational cost than bmm LoRA whenever the above inequality holds true”. Could you elaborate more about scenarios when you expect (7) to be lower than 1?
-	Please insert references to Table 1 and 2 when you comment results in Section 4.
-	Table 1: I suggest to highlight (e.g. bold text) the best improvement for each row.
-	I would move Section 5 (Related work) after the Introduction, since it provides some useful context to the presented approach.

---

> ### Author Response · Authors · 2023-11-16
>
> Thank you for your insightful review.
> > Could you explicitly clarify the definition of “expressive power” in the paper?
>
> We use "expressive power" to denote the rank of the modulation matrix. Both fLoRA and LoRA have the capability to modulate the weight matrix up to any desired rank. In this perspective, they have the same expressive power. We will make this clearer in the revision.
>
> > Expect a higher, rather than simply equivalent, accuracy compared to LoRA?
>
> In our experiments (Sec. 4.2 and Sec. 4.3), we also trained task specific LoRA heads for each language in order for a fair comparison. Hence, one wouldn’t expect fLoRA outperform LoRA in terms of accuracy on the tasks we considered. In practice, the **average task accuracy** of fLoRA might be higher LoRA assuming LoRA has to use the same adapter in a majority of samples. However, our experiments did not account for this scenario so we did not claim a higher accuracy from fLoRA.
>
> > In fLoRA the weight matrix specific for each example is the element-wise MULTIPLICATION of W0 and DeltaWi?
>
> This is correct.
>
> > Scenarios when Eq 7 is less than 1.
>
> Equation 7 will not hold true when the rank (r) is large. This is supported by the data shown in Figure 2, where LoRA's performance begins to exceed that of fLoRA as the rank increases beyond a certain threshold. We will elaborate on this in the revised version.
>
> We also appreciate the feedback on how to improve the paper’s readability such as removing references from the abstract and better framing in Fig 1. We will make these adjustments in the revision.

---

### Official Review · Reviewer_1RDh · 2023-11-02

**Soundness:** 4 excellent
**Presentation:** 3 good
**Contribution:** 3 good
**Rating:** 8
**Confidence:** 4

**Summary:**

The paper propose a new low rank adaptation technique based on a generalization of IA3.
Essentially the adaption changes from LORA: W = W0 + BA to FLORA: W = W0.*BA
This allows to pack in a batch many different adaptors per input or even per chunk efficiently.

**Strengths:**

The paper presents several strong points.
The proposed approach improves latency and throughput as well as a theoretical cost estimation.
Several model sizes from starCorder and LLama 2 are considered for throughput and latency estimation.
The accuracy of the proposed method is similar or better to that of LORA and IA3 and report improvements/checks on several models such as Llama2, whisper or starCoder.

**Weaknesses:**

The approach requires re-adapting the models that have already been adapted with LORA to leverage the improvements.
There is a breaking point where FLORA doesn't improve over LORA effectively. Intuitively, there is at least 4 factors for this: the model, the gpu architecture, the rank of the adaptation and the batch size . The rank is taken into account but it is not very clear how the other elements will come into play in practice. Eq 7 claims only important factors are the dimension of the multiplication the constants for MM and BMM and the rank. However, it is difficult to understand why this should be the case for batch size 1 in contrast to a larger batch size.
Computing some plots in this area would have been very helpful to grasp how the theoretical analysis transfer to the practical scenarios..
Another example of this would be computing per token and example adapters , which is the extreme case. It would have been interesting for latency and throughput curves to see such an extreme case, even though there is no such a real task.
The section 3.1 is confusing in its current form and a rewrite paying attention to the Matrix and elementwise operations would improve readability.
Given the constrains of the approach regarding the low-rank dimension, the applicability of the approach is limited to some specific scenarios which could have already been taken care on the base LLM pretraining. For instance, for the multilingual case the models could have already specific sparsely activated components given the language category or the programing language from the beginning.

**Questions:**

How does the batch size affects the improvements of the proposed FLORA ?
How does the picture change if we use per token and per batch adapter ?
Which other scenarios are the authors considering further from fixing lack of conditional inputs on the models ?

---

> ### Author Response · Authors · 2023-11-16
>
> Thank you for your thorough review and constructive feedback. We appreciate the opportunity to address your concerns.
>
> > Why is the batch size not appearing in Eq 7?
>
> This is because for presentation purposes we assume c1 and c2 are constants and c1 >> c2. But in the actual PyTorch implementation, c1/c2 is a function of batch size (c1 = c2 when batch size is 1 and c1>>c2 when batch size is large). Notice that in the linear layer during LLM serving, the effective batch size is # sequences * max_sequence_length so it is safe to assume c1 >> c2. We will clarify this aspect in the revision.
>
> > Throughput and latency under per token case.
>
> We appreciate your suggestion that analyzing the throughput and latency under such an extreme case will offer more insights. We agree that this analysis will provide a more comprehensive study of fLoRA's serving capabilities. This scenario is expected to demonstrate even more advantages for fLoRA over BMM LoRA. This requires a new implementation under our current modification over the vLLM framework. We will include this study in the revision. We will also consider the per batch adapter case as mentioned in your questions.
>
> > Limited applicability
>
> Thank you for pointing out the limited applicability due to the low-rank setup. We agree that some applicable scenarios can be taken care of by the base LLM pretraining. While it's true that certain scenarios could be addressed during base LLM pretraining, we want to point out that our approach is particularly beneficial for tailoring responses to specific user requests that may involve tail knowledge such as a personal codebase or personal browning history, which is usually not covered in pre-training.
>
> > Batch size
>
> Thank you for pointing out the impact of batch size. In the vLLM framework, it uses max number batched tokens to control how many tokens can be batched in one forward pass. For the throughput experiment, we set max_num_batched_tokens to be 8192, following the convention of using the model context size (Starcoder). Assuming the average input length of the testing dataset (the chat dataset to fine-tune the Vicuna model) is 512 then the batch size would be 16. For the latency experiment, we set max_num_batched_tokens to be 2560 which is the default setting of the vLLM framework for online serving.
>
> Furthermore, according to https://github.com/microsoft/DeepSpeed/blob/master/blogs/deepspeed-fastgen/README.md, section 3, the number of batched tokens is a better signal than batch size when measuring the performance in the serving scenario. The convention to measure throughput is to increase the number of batched tokens to reach the throughput-saturating regime.
>
> We ran additional experiments with max_num_batched_tokens=4096 and max_num_batched_tokens=2048. The throughput of both fLoRA and LoRA reduced roughly 10% and 20% compared to max_num_batched_tokens=8192. This is because it is not in the throughput-saturating regime of the H100 GPU. The critical rank remains the same.
>
> We will include this study in the revision.
>
> > Confusing section 3.1
>
> We appreciate your feedback on the clarity of Section 3.1. We will revise this section to improve its readability.

---

### Author Response · Authors · 2023-11-20

Dear Reviewers,

Thank you for your valuable insights and suggestions. We have tried our best to answer your questions in our author response, and we are also working to revise the paper following your suggestions. Given that the discussion period is ending soon, we were wondering if you could let us know if you have further questions or whether the author response addressed your concerns. We would be delighted to answer any further questions you might have.

Best regards,

Authors

---

### Meta-Review · Area_Chair_HYhU · 2023-12-05

**Metareview:**

The paper improves on the widely used Low-Rank Adaptation (LoRA) framework to fine-tune foundation models by introducing FLoRA ("Fast LoRA"), which allows distinct adapters (with different rank) for different task-specific requirements within the same batch. The authors empirically demonstrate that their approach preserves the advantages of LoRA in terms of accuracy on multilingual code generation and speech recognition tasks, while facilitating a higher throughput and lower latency.

Strengths:

The paper clearly motivates the problem and the experiments demonstrate improvements over the state of the art. The contribution is significant to cope with practical challenges of using foundation models in real-time serving scenarios. Reviewers agree that the paper is theoretically and technically sound and the presentation is clear, well framed in the context, and easy to follow.

Weaknesses:

Overall, weaknesses pointed out by reviewers fall either into the "minor changes" or "could state limitations more clearly" category. Reviewers point out that FLoRA requires some adjustments to the original LoRA approach (e.g. re-training/ re-adapting), so it is not a drop-in replacement and will only work for certain well-posed problems. Also, reviewers ask about the potential of outperforming LoRA, given that the additional flexibility to assign different ranks should give the model more degrees of freedom. Authors explain that this would not be the case since they accounted for that freedom in their baselines.

**Justification For Why Not Higher Score:**

n/a

**Justification For Why Not Lower Score:**

Overall, the paper has 4 clear accept recommendations, and good/ excellent scores for soundness, presentation, contribution. Author feedback should have contributed to further improvements. My only reservation would be that the paper may not be applicable to *all* work using LoRA, but from my understanding the work should still be relevant to a large part of the community and deserves visibility.

---

### Decision · Program_Chairs · 2024-01-16

Accept (oral)